# THERMOMETER ENCODING: ONE HOT WAY TO RESIST ADVERSARIAL EXAMPLES

**Jacob Buckman**[*][†]**, Aurko Roy,[*] Colin Raffel, Ian Goodfellow**
Google Brain
Mountain View, CA
{buckman, aurkor, craffel, goodfellow}@google.com

## ABSTRACT

It is well known that it is possible to construct "adversarial examples" for neural networks: inputs which are misclassified by the network yet indistinguishable from true data. We propose a simple modification to standard neural network architectures, *thermometer encoding*, which significantly increases the robustness of the network to adversarial examples. We demonstrate this robustness with experiments on the MNIST, CIFAR-10, CIFAR-100, and SVHN datasets, and show that models with thermometer-encoded inputs consistently have higher accuracy on adversarial examples, without decreasing generalization. State-of-the-art accuracy under the strongest known white-box attack was increased from 93.20% to 94.30% on MNIST and 50.00% to 79.16% on CIFAR-10. We explore the properties of these networks, providing evidence that thermometer encodings help neural networks to find more-non-linear decision boundaries.

## 1 INTRODUCTION AND RELATED WORK

*Adversarial examples* are inputs to machine learning models that are intentionally designed to cause the model to produce an incorrect output. The term was introduced by Szegedy et al. (2014) in the context of neural networks for computer vision. In the context of spam and malware detection, such inputs have been studied earlier under the name *evasion attacks* (Biggio et al., 2013). Adversarial examples are interesting from a scientific perspective, because they demonstrate that even machine learning models that have superhuman performance on I.I.D. test sets fail catastrophically on inputs that are modified even slightly by an adversary. Adversarial examples also raise concerns in the emerging field of *machine learning security* because malicious attackers could use adversarial examples to cause undesired behavior (Papernot et al., 2016).

Unfortunately, there is not yet any known strong defense against adversarial examples. Adversarial examples that fool one model often fool another model, even if the two models are trained on different training examples (corresponding to the same task) or have different architectures (Szegedy et al., 2014), so an attacker can fool a model without access to it. Attackers can improve their success rate by sending inputs to a model, observing its output, and fitting their own own copy of the model to the observed input-output pairs (Papernot et al., 2016). Attackers can also improve their success rate by searching for adversarial examples that fool multiple different models—such adversarial examples are then much more likely to fool the unknown target model (Liu et al., 2016). Szegedy et al. (2014) proposed to defend the model using *adversarial training* (training on adversarial examples as well as regular examples) but it was not feasible to generate enough adversarial examples in the inner loop of the training process for the method to be effective at the time. Szegedy et al. (2014) used a large number of iterations of L-BFGS to produce their adversarial examples. Goodfellow et al. (2014) developed the *fast gradient sign method* (FGSM) of generating adversarial examples and demonstrated that adversarial training is effective for reducing the error rate on adversarial examples. A major difficulty of adversarial training is that it tends to overfit to the method of adversarial example generation used at training time. For example, models trained to resist FGSM adversarial examples usually fail to resist L-BFGS adversarial examples. Kurakin et al. (2016) introduced the

---

[*]Equal contribution.
[†]Work done as a member of the Google AI Residency program (g.co/airesidency)

*basic iterative method* (BIM) which lies between FGSM and L-BFGS on a curve trading speed for effectiveness (the BIM consists of running FGSM for a medium number of iterations). Adversarial training using BIM still overfits to the BIM, unfortunately, and different iterative methods can still successfully attack the model. Recently, Madry et al. (2017) showed that adversarial training using adversarial examples created by adding random noise before running BIM results in a model that is highly robust against all known attacks on the MNIST dataset. However, it is less effective on more complex datasets, such as CIFAR. A strategy for training networks which are robust to adversarial attacks across all contexts is still unknown. In this work, we demonstrate that thermometer code discretization and one-hot code discretization of real-valued inputs to a model significantly improves its robustness to adversarial attack, advancing the state of the art in this field.

## 2 INPUT DISCRETIZATION

We propose to break the linear extrapolation behavior of machine learning models by preprocessing the input with an extremely nonlinear function. This function must still permit the machine learning model to function successfully on naturally occurring inputs. The recent success of the PixelRNN model (Oord et al., 2016) has demonstrated that one-hot discrete codes for 256 possible values of color pixels are effective representations for input data. Other extremely nonlinear functions may also defend against adversarial examples, but we focused attention on vector-valued discrete encoding as our nonlinear function because of the evidence from PixelRNN that it would support successful machine learning.

Images are often encoded as a 3D tensor of integers in the range [0, 255]. The tensor's three dimensions correspond to the image's height, width, and color channels (e.g. three for RGB, one for greyscale). Each value represents an intensity value for a given color at a given horizontal/vertical position. For classification tasks, these values are typically normalized to floating-point approximations in the range (0, 1). Input discretization refers to the process of separating these continuous-valued pixel inputs into a set of non-overlapping *buckets*, which are each mapped to a fixed binary vector.

Past work, for example *depth-color-squeezing* (Xu et al., 2017), has explored what we will refer to as *quantization* of inputs as a potential defense against adversarial examples. In that approach, each pixel value is mapped to the low-bit version of its original value, which is a fixed scalar. The key novel aspect of our approach is that rather than replacing a real number with a number of low bit depth, we replace each real number with a binary vector. Different values of the real number activate different bits of the input vector. Multiplying the input vector by the network's weights thus performs an operation similar to an embedding lookup in a language model, so different input values actually use different parameters of the network. To avoid confusion, we will consistently refer to scalar-to-scalar precision reduction as *quantization* and scalar-to-vector encoding schemes as *discretization* throughout this work. A comparison of these techniques can be seen in Table 1. Note that, unlike depth-color-squeezing, discretization makes a meaningful change to the model, even when it is configured to use enough discretization levels to avoid losing any information from a traditionally formatted computer vision training set; discretizing each pixel to 256 levels will preserve all of the information contained in the original image. Discretization defends against adversarial examples by changing which parameters of the model are used, and may also discard information if the number of discretization levels is low; quantization can only defend the model by discarding information.

| Real-valued | Quantized | Discretized (one-hot) | Discretized (thermometer) |
|:---:|:---:|:---:|:---:|
| 0.13 | 0.15 | [0100000000] | [0111111111] |
| 0.66 | 0.65 | [0000001000] | [0000001111] |
| 0.92 | 0.95 | [0000000001] | [0000000001] |

Table 1: Examples mapping from continuous-valued inputs to quantized inputs, one-hot codes, and thermometer codes, with ten evenly-spaced levels.

### 2.1 DISCRETIZATION AS A DEFENSE

In Goodfellow et al. (2014), the authors provide evidence that several network architectures, including LSTMs (Hochreiter & Schmidhuber, 1997), sigmoid networks (Han & Moraga, 1995), and

maxout networks (Goodfellow et al., 2013), are vulnerable to adversarial examples due to the empirical fact that, when trained, the loss function of these networks tends to be highly linear with respect to its inputs.

We briefly recall the reasoning of Goodfellow et al. (2014). Assume that we have a logistic regressor with weight matrix $w$. Consider an image $x \in \mathbb{R}^n$ which is perturbed into $\widetilde{x} = x + \eta$ by some noise $\eta$ such that $\|\eta\|_\infty \leq \varepsilon$ for some $\varepsilon$. The probability that the model assigns to the true class is equal to:

$$\mathbb{L}(\widetilde{x}) = \sigma(w^\top \widetilde{x}) = \sigma(w^\top (x + \eta)) = \sigma(w^\top x + w^\top \eta)$$

If the perturbation $\eta$ is adversarial, such as in the case where $\eta_i := \varepsilon \cdot \text{sign}\left(\frac{\partial \mathbb{L}(x)}{\partial x_i}\right)$, then the input to the sigmoid is increased by $\varepsilon \cdot n$. If $n$ is large, as is typically the case in images and other high-dimensional spaces of interest, this linearity implies that even imperceptibly small values of $\varepsilon$ can have a large effect on the model's prediction, making the model vulnerable to adversarial attacks.

Though neural networks in principle have the capacity to represent highly nonlinear functions, networks trained via stochastic gradient descent on real-world datasets tend to converge to mostly-linear solutions. This is illustrated in the empirical studies conducted by Goodfellow et al. (2014). One hypothesis proposed to explain this phenomenon is that the nonlinearities typically used in networks are either piecewise linear, like ReLUs, or approximately linear in the parts of their domain in which training takes place, like the sigmoid function.

One potential solution to this problem is to use more non-linear activation functions, such as quadratic or RBF units. Indeed, it was shown by Goodfellow et al. (2014) that such units were more resistant to adversarial perturbations of their inputs. However, these units are difficult to train, and the resulting models do not generalize very well (Goodfellow et al., 2014), sacrificing accuracy on clean examples. As an alternative to introducing highly non-linear activation functions in the network, we propose applying a non-differentiable and non-linear transformation (*discretization*) to the input, before passing it into the model. A comparison of the input to the model under various regimes can be seen in Figure 1, highlighting the strong non-linearity of discretization techniques.

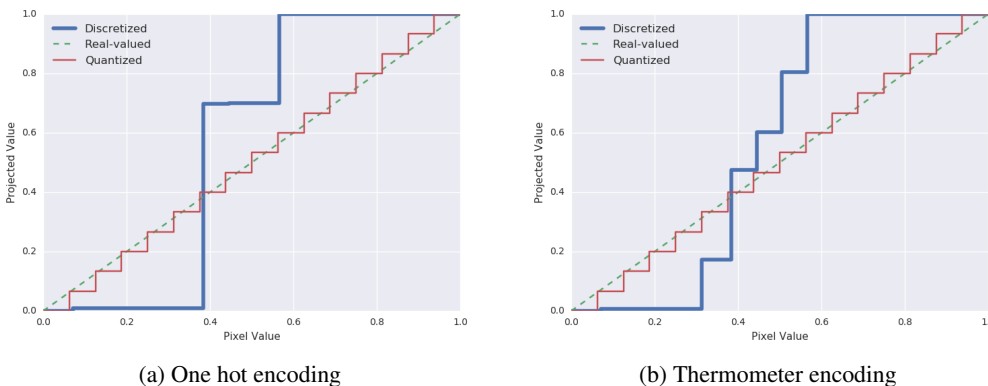

(a) One hot encoding             (b) Thermometer encoding

Figure 1: Comparison of regular inputs, quantized inputs, and discretized inputs (16 levels, projected to one dimension) on MNIST, adversarially trained with $\varepsilon = 0.3$. The $x$-axis represents the true pixel value of the image, and the $y$-axis represents the value that is passed as input to the network after the input transformation has been applied. For real-valued inputs, the inputs to the network are affected linearly by perturbations to the input. Quantized inputs are also affected approximately linearly by perturbations where $\varepsilon$ is greater than the bucket width. Discretizing the input, and then using learned weights to project the discretized value back to a single scalar, we see that the model has learned a highly non-linear function to represent the input in a fashion that is effective for resisting the adversarial perturbations it has seen. When starting at the most common pixel-values for MNIST, 0 and 1, any perturbation of the pixels (where $\varepsilon \leq 0.3$) has barely any effect on the input to the network.

## 2.2 TYPES OF DISCRETIZATION

In this work we consider two approaches to constructing discretized representations $f(x)$ of the input image $x$. Assume for the sake of simplicity that the entries of $x$ take values in the continuous domain $[0, 1]$.

We first describe a quantization function $b$. Choose $0 < b_1 < b_2 < \cdots < b_k = 1$ in some fashion. (In this work, we simply divide the domain evenly, i.e. $b_i = \frac{i}{k}$.) For a real number $\theta \in [0, 1]$ define $b(\theta)$ to be the largest index $\alpha \in \{1, \ldots, k\}$ such that $\theta \leq b_\alpha$.

### 2.2.1 ONE-HOT ENCODINGS

For an index $j \in \{1, \ldots, k\}$ let $\chi(j) \in \mathbb{R}^k$ be the indicator or one-hot vector of $j$, i.e.,

$$\chi(j)_l = \begin{cases} 1 & \text{if } l = j \\ 0 & \text{otherwise.} \end{cases}$$

The discretization function is defined pixel-wise for a pixel $i \in \{1, \ldots, n\}$ as:

$$f_{onehot}(x_i) = \chi\left(b(x_i)\right).$$

One-hot encodings are simple to compute and understand, and are often used when it is necessary to represent a categorical variable in a neural network. However, one-hot encodings are not well suited for representing categorical variables with an interpretation of ordering between them. Note that the ordering information between two pixels $x_i$ and $x_j$ is lost by applying the transformation $f_{onehot}$; for a pair of pixels $i, j$ whenever $b(x_i) \neq b(x_j)$, we see:

$$\|\chi(b(x_i))\|_2 = \|\chi(b(x_j))\|_2 = 1.$$

In the case of pixel values, this is not a good inductive bias, as there is a clear reason to believe that neighboring buckets are more similar to each other than distant buckets.

### 2.2.2 THERMOMETER ENCODINGS

In order to discretize the input image $x$ without losing the relative distance information, we propose *thermometer encodings*. For an index $j \in \{1, \ldots, k\}$, let $\tau(j) \in \mathbb{R}^k$ be the thermometer vector defined as

$$\tau(j)_l = \begin{cases} 1 & \text{if } l \geq j \\ 0 & \text{otherwise.} \end{cases}$$

Then the discretization function $f$ is defined pixel-wise for a pixel $i \in \{1, \ldots, n\}$ as:

$$f_{therm}(x)_i = \tau\left(b(x_i)\right) = \mathbb{C}(f_{onehot}(x_i))$$

where $\mathbb{C}$ is the cumulative sum function, $\mathbb{C}(c)_l = \sum_{j=0}^{l} c_l$.

Note that the thermometer encoding preserves pairwise ordering information, i.e., for pixels $i, j$ if $b(x_j) \neq b(x_k)$ and $x_i < x_j$ then

$$\|\tau(b(x_i))\|_2 < \|\tau(b(x_j))\|_2.$$

## 2.3 WHITE-BOX ATTACKS ON DISCRETIZED INPUTS

Discretizing the input makes it difficult to attack the model with standard white-box attack algorithms, such as FGSM (Goodfellow et al., 2014) and PGD (Madry et al., 2017), since it is impossible to backpropagate through our discretization function to determine how to adversarially modify the model's input. In this section, we describe two novel iterative attacks which allow us to construct adversarial examples for networks trained on discretized inputs.

Constructing white-box attacks on discretized inputs serves two primary purposes. First, it allows us to more completely evaluate whether the model is robust to all adversarial attacks, as white-box attacks are typically more powerful than their black-box counterparts. Secondly, adversarial training

is typically performed in a white-box fashion, and so in order to utilize and properly compare against the adversarial training techniques of Madry et al. (2017), it is important to have strong white-box attacks.

For ease of presentation, we will describe the attacks assuming that $f : \mathbb{R} \to \mathbb{R}^k$ discretizes inputs into thermometer encodings; in order to attack one-hot encodings, simply replace all instances of $f_{therm}$ with $f_{onehot}$, $\tau$ with $\chi$, and $\mathbb{C}$ with the identity function $\mathbb{I}$. We represent the adversarial image after $t$ steps of the attack as $z^t$, where the value of the $i$th pixel is $z_i^t$.

The first attack, Discrete Gradient Ascent (DGA), follows the direction of the gradient of the loss with respect to $f(x)$, but is constrained at every step to be a discretized vector. If we have discretized the input image into $k$-dimensional vectors using the one-hot encoding, this corresponds to moving to a vertex of the simplex $(\Delta_k)^n$ at every step. The second attack, Logit-Space Projected Gradient Ascent (LS-PGA), relaxes this assumption, allowing intermediate iterates to be in the interior of the simplex. The final adversarial image is obtained by projecting the final point back to the nearest vertex of the simplex.

Note that if the number of attack steps is 1, then the two attacks are equivalent; however, for larger numbers of attack steps, LS-PGA is a generalization of DGA.

### 2.3.1 DISCRETE GRADIENT ASCENT (DGA)

Following PGD (Madry et al., 2017), we initialize DGA by placing each pixel into a random bucket that is within $\varepsilon$ of the pixel's true value. At each step of the attack, we look at all buckets that are within $\varepsilon$ of the true value, and select the bucket that is likely to do the most 'harm', as estimated by the gradient of setting that bucket's indicator variable to 1, with respect to the model's loss at the previous step.

$$z_i^0 = f_{therm}(x_i + U(-\varepsilon, \varepsilon))$$

$$\text{harm}(z_i^t)_l = \begin{cases} (z_i^t - \tau(l))^\top \cdot \frac{\partial \mathbb{L}(z^t)}{\partial z_i^t} & \text{if } \exists(-\varepsilon \leq \eta \leq \varepsilon) \quad \text{s.t.} \quad b(x_i + \eta) = l \\ 0 & \text{otherwise.} \end{cases}$$

$$z_i^{t+1} = \tau\left(\arg\max\left(\text{harm}\left(z_i^t\right)\right)\right)$$

Because the outcome of this optimization procedure will vary depending on the initial random perturbation, we suggest strengthening the attack by re-running it several times and using the perturbation with the greatest loss. The pseudo-code for the DGA attack is given in Section B of the appendix.

### 2.3.2 LOGIT-SPACE PROJECTED GRADIENT ASCENT (LS-PGA)

To perform LS-PGA, we soften the discrete encodings into continuous relaxations, and then perform standard Projected Gradient Ascent (PGA) on these relaxed values. We represent the distribution over embeddings as a softmax over logits $u$, each corresponding to the unnormalized log-weight of a specific bucket's embedding. To improve the attack, we scale the logits with temperature $T$, allowing us to trade off between how closely our softmax approximates a true one-hot distribution as in the Gumbel-softmax trick (Jang et al., 2016; Maddison et al., 2016), and how much gradient signal the logits receive. At each step of a multi-step attack, we anneal this value via exponential decay with rate $\delta$.

$$z_i^t = \mathbb{C}\left(\sigma\left(\frac{u_i^t}{T^t}\right)\right)$$

$$z_i^{final} = \tau\left(\arg\max\left(u_i^{final}\right)\right)$$

$$T^t = T^{t-1} \cdot \delta$$

We initialize each of the logits randomly with values sampled from a standard normal distribution. At each step, we ensure that the model does not assign any probability to buckets which are not within $\varepsilon$ of the true value by fixing the logits to be $-\infty$. The model's loss is a continuous function of the logits, so we can simply utilize attacks designed for continuous-valued inputs, in this case PGA

with step-size $\xi$.

$$u_i^0 = \begin{cases} \mathcal{N}\left(\mathbf{0}; \mathbf{1}\right) & \text{if } \exists(-\varepsilon \leq \eta \leq \varepsilon) \quad \text{s.t.} \quad b(x_i + \eta) = l \\ -\infty & \text{otherwise.} \end{cases}$$

$$\left(u_i^{t+1}\right)_l = \begin{cases} (u_i^t)_l + \xi \cdot \text{sign}\left(\frac{\partial \mathbb{L}(z^t)}{\partial u_i^t}\right)_l & \text{if } \exists(-\varepsilon \leq \eta \leq \varepsilon) \quad \text{s.t.} \quad b(x_i + \eta) = l \\ -\infty & \text{otherwise.} \end{cases}$$

Because the outcome of this optimization procedure will vary depending on the initial perturbation, we suggest strengthening the attack by re-running it several times and using the perturbation with the greatest loss. The pseudo-code for the LS-PGA attack is given in Section B of the appendix.

## 3 EXPERIMENTS

We compare models trained with input discretization to state-of-the-art adversarial defenses on a variety of datasets. We match the experimental setup of the prior literature as closely as possible. Rows labeled with "Vanilla (Madry)" give the numbers reported in Madry et al. (2017); other rows contain results of our own experiments, with "Vanilla" containing a direct replication. For our MNIST experiments, we use a convolutional network; for CIFAR-10, CIFAR-100, and SVHN we use a Wide ResNet (Zagoruyko & Komodakis, 2016). We use a network of depth 30 for the CIFAR-10 and CIFAR-100 datasets, while for SVHN we use a network of depth 15. The width factor of all the Wide ResNets is set to $k = 4$. [1] Unless otherwise specified, all quantized and discretized models use 16 levels.

We found that in all cases, LS-PGA was strictly more powerful than DGA, so all attacks on discretized models use LS-PGA with $\xi = 0.01, \delta = 1.2$, and 1 random restart. To be consistent with Madry et al. (2017), we describe attacks in terms of the maximum $\ell_\infty$-norm of the attack, $\varepsilon$. All MNIST experiments used $\varepsilon = 0.3$ and 40 steps for iterative attacks; experiments on CIFAR used $\varepsilon = 0.031$ and 7 steps for iterative attacks; experiments on SVHN used $\varepsilon = 0.047$ and 10 steps for iterative attacks. These settings were used for adversarial training, white-box attacks, and black-box attacks. Figure 3 plots the effectiveness of the iterated PGD/LS-PGA attacks on vanilla and discretized models for MNIST and shows that increasing the number of iterations beyond 40 would have no effect on the performance of the model on $\ell_\infty$-bounded adversarial examples for MNIST.

In Madry et al. (2017), adversarially-trained models are trained using exclusively adversarial inputs. This led to a small but noticeable loss in accuracy on clean examples, dropping from 99.2% to 98.8% on MNIST and from 95.2% to 87.3% on CIFAR-10 in return for more robustness towards adversarial examples. Past work has also sometimes performed adversarial training on batches composed of half clean examples and half adversarial examples (Goodfellow et al., 2014; Cisse et al., 2017). To be consistent with Madry et al. (2017), we list experiments on models trained *only* on adversarial inputs in the main paper; additional experiments on a *mix* of clean and adversarial inputs can be found in the appendix.

We also run experiments exploring the model's relationship with the number of distinct levels to which we quantize the input before discretizing it, and exploring various settings of hyperparameters for LS-PGA.

## 4 RESULTS

Our adversarially-trained baseline models were able to approximately replicate the results of Madry et al. (2017). On all datasets, discretizing the inputs of the network dramatically improves resistance to adversarial examples, while barely sacrificing any accuracy on clean examples. Quantized models also beat the baseline, but with lower accuracy on clean examples. Discretization via thermometer encodings outperformed one-hot encodings in most settings. See Tables 2,3,4 and 5 for results on MNIST and CIFAR-10. Additional results on CIFAR-100 and SVHN are included in the appendix.

---

[1]A full list of hyperparameters can be found in the appendix. Source code is available at http://anonymized

In Figures 2 and 5 (located in appendix), we plot the test-set accuracy across training timesteps for various adversarially trained models on the SVHN and CIFAR-10 datasets, and observe that the discretized models become robust against adversarial examples more quickly.

| | Model | Clean | FGSM | PGD/LS-PGA |
|---|---|---|---|---|
| *Clean* | Vanilla (Madry) | 99.20 | 6.40 | - |
| | Vanilla | **99.30** | 0.19 | 0 |
| | Quantized | 99.19 | 1.10 | 0 |
| | One-hot | 99.13 | 0 | 0 |
| | Thermometer | 99.20 | 0 | 0 |
| *Adv. train* | Vanilla (Madry) | 98.80 | 95.60 | 93.20 |
| | Vanilla | 98.67 | 96.17 | 93.30 |
| | Quantized | 98.75 | **96.29** | 94.23 |
| | One-hot | 98.61 | 96.22 | **94.30** |
| | Thermometer | 99.03 | 95.84 | 94.02 |

Table 2: Comparison of adversarial robustness to *white-box attacks* on MNIST .

| | Source / Target | Clean | | | Adv. train | | |
|---|---|---|---|---|---|---|---|
| | | **Vanilla** | **One-hot** | **Thermometer** | **Vanilla** | **One-hot** | **Thermometer** |
| *Clean* | Vanilla | 2.04 | 36.02 | 24.58 | 3.48 | 80.44 | 57.69 |
| | Quantized | 39.22 | 32.39 | 25.63 | 75.02 | 75.92 | 52.32 |
| | One-hot | 14.57 | 6.91 | 8.11 | 39.02 | 39.60 | 18.02 |
| | Thermometer | 41.12 | 14.30 | 10.98 | 61.84 | 59.16 | 32.93 |
| *Adv. train* | Vanilla (Madry) | - | - | - | 96.0 | - | - |
| | Quantized | 97.65 | 98.16 | 97.14 | 95.27 | 95.31 | 96.53 |
| | Vanilla | 97.62 | 98.05 | 97.06 | 95.43 | 95.38 | 96.23 |
| | One-hot | 97.78 | 98.48 | 97.87 | 96.87 | 96.60 | 96.87 |
| | Thermometer | **98.07** | **98.75** | **98.02** | **97.05** | **96.88** | **97.13** |

Table 3: Comparison of adversarial robustness to *black-box attacks* on MNIST .

| | Model | Clean | FGSM | PGD/LS-PGA |
|---|---|---|---|---|
| *Clean* | Vanilla (Madry) | **95.20** | 25.10 | 4.10 |
| | Vanilla | 94.29 | 46.15 | 1.66 |
| | Quantized | 93.49 | 43.89 | 3.57 |
| | One-hot | 93.26 | 52.07 | 53.11 |
| | Thermometer | 94.22 | 48.50 | 50.50 |
| *Adv. train* | Vanilla (Madry) | 87.3 | 60.3 | 50.0 |
| | Vanilla | 87.67 | 59.7 | 41.78 |
| | Quantized | 85.75 | 53.53 | 42.09 |
| | One-hot | 88.67 | 68.76 | 67.83 |
| | Thermometer | 89.88 | **80.96** | **79.16** |

Table 4: Comparison of adversarial robustness to *white-box attacks* on CIFAR-10 .

## 5 DISCUSSION

In Goodfellow et al. (2014), the seeming linearity of deep neural networks was shown by visualizing the networks in several different ways. To test our hypothesis that discretization breaks some of this linearity, we replicate these visualizations and contrast them to visualizations of discretized models. See Appendix G for an illustration of these properties.

For non-discretized, clean trained models, test-set examples always yield a linear boundary between correct and incorrect classification; in contrast, non-adversarially-trained models have a more interesting *parabolic shape* (see Figure 9).

| Source | | Clean | | | Adv. train | | |
|---|---|---|---|---|---|---|---|
| Target | | Vanilla | One-hot | Thermometer | Vanilla | One-hot | Thermometer |
| *Clean* | Vanilla (Madry) | 0.0 | - | - | 79.7 | - | - |
| | Vanilla | 3.38 | 60.10 | 52.60 | 45.48 | 37.21 | 49.91 |
| | Quantized | 70.54 | 62.46 | 55.38 | 51.74 | 45.37 | 55.64 |
| | One-hot | 83.00 | 56.25 | 63.94 | 54.59 | 49.21 | 57.28 |
| | Thermometer | 80.33 | 66.22 | 53.45 | 57.04 | 51.03 | 60.90 |
| *Adv. train* | Vanilla (Madry) | 85.60 | - | - | 67.0 | - | - |
| | Vanilla | 85.60 | 74.99 | 73.78 | 67.0 | 50.09 | 71.03 |
| | Quantized | 84.56 | **82.43** | **82.22** | **72.52** | **72.29** | **79.43** |
| | One-hot | 86.01 | 77.19 | 77.70 | 61.92 | 60.02 | 72.89 |
| | Thermometer | **88.25** | 81.59 | 80.80 | 67.96 | 67.43 | 77.68 |

Table 5: Comparison of adversarial robustness to *black-box attacks* on CIFAR-10 .

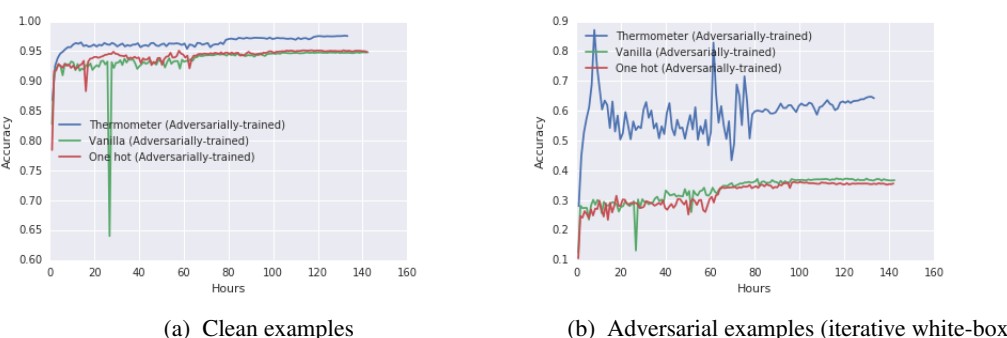

(a) Clean examples          (b) Adversarial examples (iterative white-box)

Figure 2: Comparison of the convergence rate of various *adversarially trained* models on the SVHN dataset.

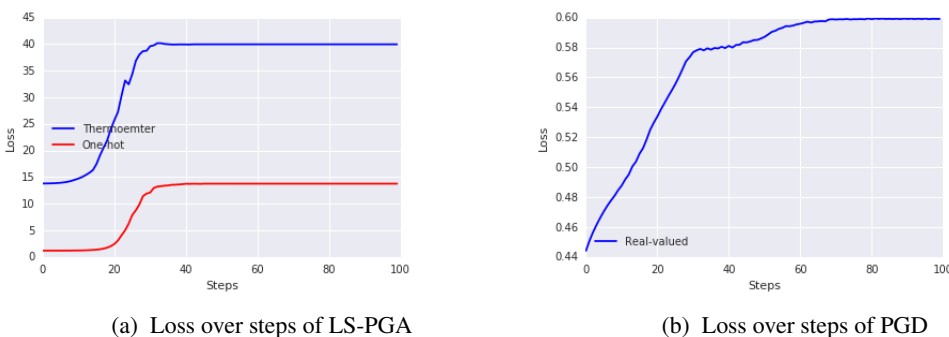

(a) Loss over steps of LS-PGA          (b) Loss over steps of PGD

Figure 3: Loss for iterated white-box attacks on various models on a randomly chosen data point from MNIST. By step 40, which is where we evaluate, the loss of the point found by iterative attacks has converged.

When discretizing the input, we introduce $C_w \cdot C_h \cdot C_o \cdot c \cdot (k - 1)$ extra parameters, where $c$ is the number of channels in the image, $k$ is the number of levels of discretization, and $C_w, C_h, C_o$ are the width, height, and output channels of the first convolutional layer. Discretizing using 16 levels introduced $0.03\%$ extra parameters for MNIST, $0.08\%$ for CIFAR-10 and CIFAR-100, and $2.3\%$ for SVHN. This increase is negligible, so it is likely that the robustness comes from the input discretization, and is not merely a byproduct of having a slightly higher-capacity model.

## 6 CONCLUSION

Our findings convincingly demonstrate that the use of thermometer encodings, in combination with adversarial training, can reduce the vulnerability of neural network models to adversarial attacks. Our analysis reveals that the resulting networks are significantly less linear with respect to their inputs, supporting the hypothesis of Goodfellow et al. (2014) that many adversarial examples are caused by over-generalization in networks that are too linear.

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

## A   HYPERPARAMETERS

In this section, we describe the hyperparameters used in our experiments. For CIFAR-10 and CIFAR-100 we follow the standard data augmenting scheme as in (Lin et al., 2013; He et al., 2016; Huang et al., 2016; Zagoruyko & Komodakis, 2016): each training image is zero-padded with $4$ pixels on each side and randomly cropped to a new $32 \times 32$ image. The resulting image is randomly flipped with probability $0.5$, it's brightness is adjusted with a delta chosen uniformly at random in the interval $[-63, 63)$ and it's contrast is adjusted using a random contrast factor in the interval $[0.2, 1.8]$. For MNIST we use the Adam optimizer with a fixed learning rate of $1e-4$ as in Madry et al. (2017). For CIFAR-10 and CIFAR-100 we use the Momentum optimizer with momentum $0.9$, $\ell_2$ weight decay of $\lambda = 0.0005$ and an initial learning rate of $0.1$ which is annealed by a factor of $0.2$ after epochs $60, 120$ and $160$ respectively as in Zagoruyko & Komodakis (2016). For SVHN we use the same optimizer with initial learning rate of $1e-2$ which is annealed by a factor of $0.1$ after epochs $80$ and $120$ respectively. We also use a dropout of $0.3$ for CIFAR-10, CIFAR-100 and SVHN.

## B   PSEUDO-CODE

The DGA attack is described in Algorithm 2 and the LS-PGA attack is described in Algorithm 3. Both these algorithms make use of a $getMask()$ sub-routine which is described in Algorithm 1.

---

**Input:** Image $x$, parameter $\varepsilon$
**Output:** $\varepsilon$-discretized mask around $x$
1  mask $\leftarrow (0)^{n \times k}$
2  low $\leftarrow \max\{0, x - \varepsilon\}$
3  high $\leftarrow \min\{1, x + \varepsilon\}$
4  **for** $\alpha \leftarrow 0$ **to** $1$ **by** $\frac{1}{k}$ **do**
5  $\quad |$ mask $\leftarrow$ mask $+ f_{onehot}(\alpha * \text{low} + (1 - \alpha) * \text{high})$
6  **end**
7  **return** mask

**Algorithm 1:** Sub-routine for getting an $\varepsilon$-discretized mask of an image.

---

## C   ADDITIONAL EXPERIMENTS ON MNIST

In this section we list the additional experiments we performed using discretized models on MNIST. The main hyperparameters of Algorithm 3 are the step size $\xi$ used to perform the projected gradient ascent, and the annealing rate of $\delta$. We found that the choice of these hyperparameters was not critical to the robustness of the model. In particular, we performed experiments with $\xi = 1.0$ and $\xi = 0.001$, and both achieved similar accuracies as in Table 2 and Table 3. Additionally, we found that without annealing, i.e., $\delta = 1.0$, the performance was only slightly worse than with $\delta = 1.2$.

We also experimented with discretizing by using *percentile information* per color channel instead of using *uniformly distributed buckets*. This did not result in any significant changes in robustness or accuracy for the MNIST dataset.

**Input:** Image $x$, label $y$, discretization function $f$, loss $\mathbb{L}(\boldsymbol{\theta}, f(x), y)$, $l$ attack steps, parameter $\varepsilon$
**Output:** Adversarial input to the network $z'$
1   $\eta \leftarrow U(-\varepsilon, \varepsilon)$
2   $z_0 \leftarrow f(x + \eta)$
3   $\text{mask} \leftarrow getMask(x, \varepsilon)$
4   **for** $t \leftarrow 1$ **to** $l$ **do**
     /* Loop invariant: $z_i^t$ is discretized for every pixel $i$     */
5     $\text{grad} \leftarrow \nabla_{z^{t-1}} \mathbb{L}(\boldsymbol{\theta}, z^{t-1}, y)$
6     **if** $f$ *is one-hot* **then**
7       $\text{harm}_l \leftarrow (z^{t-1} - \chi(l))^\top \text{grad}$
8     **else**
9       $\text{harm}_l \leftarrow (z^{t-1} - \tau(l))^\top \text{grad}$
10    **end**
11    $\text{harm} \leftarrow \text{harm} * \text{mask} - (1 - \text{mask}) * \infty$
12    $z^t \leftarrow f(\arg\max(\text{harm}))$
13   **end**
14   **return** $z' \leftarrow z^l$

**Algorithm 2:** Discrete Gradient Ascent (DGA)

**Input:** Image $x$, label $y$, discretization function $f$, loss $\mathbb{L}(\boldsymbol{\theta}, f(x), y)$, $l$ attack steps, parameters $\varepsilon, \delta$
**Output:** Adversarial input to the network $z'$
1   $\text{mask} \leftarrow getMask(x, \varepsilon)$
2   $u^0 \leftarrow \left(\mathcal{N}(\mathbf{0}^k; \mathbf{1}^k)\right)^n * \text{mask} - (1 - \text{mask}) * \infty$
3   $T \leftarrow 1$
4   **if** $f$ *is one-hot* **then**
5     $\mathbb{F} \leftarrow \mathbb{I}$
6   **else**
7     $\mathbb{F} \leftarrow \mathbb{C}$
8   **end**
9   $z^0 \leftarrow \mathbb{F}\left(\sigma\left(\frac{u^0}{T}\right)\right)$
10   **for** $t \leftarrow 1$ **to** $l$ **do**
11    $\text{grad} \leftarrow \nabla_{u^{t-1}} \mathbb{L}(\boldsymbol{\theta}, z^{t-1}, y)$
12    $u^t \leftarrow u^{t-1} + \xi \cdot \text{sign}(\text{grad})$
13    $z^t \leftarrow \mathbb{F}\left(\sigma\left(\frac{u^t}{T}\right)\right)$
14    $T \leftarrow T \cdot \delta$
15   **end**
16   **return** $z' \leftarrow \tau(\arg\max(u_i^l))$

**Algorithm 3:** Logit-Space Projected Gradient Ascent (LS-PGA)

| Model | $\xi$ | $\delta$ | White-box | Black-box |
|---|---|---|---|---|
| One-hot | 1.0 | 1.2 | 93.08 | 98.39 |
| One-hot | 0.001 | 1.2 | 93.50 | 98.42 |
| One-hot | 1.0 | 1.0 | 93.02 | 98.21 |
| Thermometer | 1.0 | 1.2 | 93.74 | **98.45** |
| Thermometer | 0.001 | 1.2 | **93.88** | 98.24 |
| Thermometer | 1.0 | 1.0 | 93.75 | 98.22 |

Table 6: Comparison of adversarial robustness to *white-box attacks* on MNIST using 16 levels and with various choices of the hyperparameters $\xi$ and $\delta$ for Algorithm 3. The models are evaluated on white-box attacks and on black-box attacks using a vanilla, clean trained model; both use LS-PGA.

Finally, we also trained on a *mix* of clean and adversarial examples: this resulted in significantly higher accuracy on clean examples, but decreased accuracy on white-box and black-box attacks compared to Tables 2 and 3.

| Model | Clean | FGSM | PGD/LS-PGA |
|---|---|---|---|
| Vanilla | 99.03 | 95.70 | 91.36 |
| One-hot | 99.01 | **96.14** | **93.77** |
| Thermometer | **99.13** | 96.10 | 93.70 |

Table 7: Comparison of adversarial robustness to *white-box attacks* on MNIST using a *mix* of clean and adversarial examples.

| Source
Target | Clean | | | Adv. train | | |
|---|---|---|---|---|---|---|
| | **Vanilla** | **One-hot** | **Thermometer** | **Vanilla** | **One-hot** | **Thermometer** |
| Vanilla | 97.88 | 97.87 | 96.99 | 93.07 | 90.97 | 96.46 |
| One-hot | 98.28 | **98.83** | **98.08** | 95.73 | **95.96** | **97.25** |
| Thermometer | **98.45** | 98.70 | 98.08 | **96.35** | 95.72 | 96.97 |

Table 8: Comparison of adversarial robustness to *black-box attacks* on MNIST of various models using a *mix* of clean and adversarial examples.

| Levels | Clean | White-box
PGD/LS-PGA | Black-box
Vanilla, Clean | Black-box
Vanilla, PGD |
|---|---|---|---|---|
| One-hot (4) | 99.09 | 92.95 | 98.23 | 95.64 |
| One-hot (8) | 99.10 | 92.65 | **98.49** | **96.37** |
| One-hot (16) | **99.14** | 93.08 | 98.39 | 96.26 |
| One-hot (32) | 99.06 | 93.51 | 98.38 | 95.78 |
| One-hot (64) | 98.89 | 93.63 | 98.35 | 95.74 |
| Thermometer (4) | 99.11 | 92.62 | 98.23 | 95.67 |
| Thermometer (8) | 99.08 | 93.45 | 98.44 | 95.93 |
| Thermometer (16) | 99.07 | 93.88 | 98.24 | 95.28 |
| Thermometer (32) | 99.02 | 94.14 | 98.24 | 95.49 |
| Thermometer (64) | 99.00 | **94.62** | 98.33 | 95.71 |

Table 9: Comparison of adversarial robustness on MNIST as the number of levels of discretization is varied. All models are trained *mix* of adversarial examples and clean examples.

## D   ADDITIONAL EXPERIMENTS ON CIFAR-10

In this section we list the additional experiments we performed on CIFAR-10. Firstly, we trained models on a *mix* of both clean and adversarial examples. The results for mixed training are listed in Tables 10 and 11; as expected it has lower accuracy on adversarial examples, but higher accuracy on clean examples, compared to training on only adversarial examples (Tables 4 and 5).

| Model | Clean | FGSM | PGD/LS-PGA |
|---|---|---|---|
| Vanilla | 87.16 | 54.50 | 34.71 |
| One-hot | 92.19 | 58.87 | 58.96 |
| Thermometer | 92.32 | **66.60** | **65.67** |

Table 10: Comparison of adversarial robustness to *white-box attacks* on CIFAR-10 of various models using a *mix* of regular and adversarial training.

| Source / Target | Clean | | | Adv. train | | |
|---|---|---|---|---|---|---|
| | **Vanilla** | **One-hot** | **Thermometer** | **Vanilla** | **One-hot** | **Thermometer** |
| Vanilla | 83.62 | 82.06 | 74.31 | 52.83 | 58.73 | 65.24 |
| One-hot | **88.16** | 75.11 | **75.49** | 61.89 | **59.17** | 69.73 |
| Thermometer | 87.50 | **75.91** | 74.84 | 61.39 | 58.51 | 68.22 |

Table 11: Comparison of adversarial robustness to *black-box attacks* on CIFAR-10 of various models using a *mix* of clean and adversarial examples.

In order to explore whether the number of levels of discretization affected the performance of the model, we trained several models which varied this number. As expected, we found that models with fewer levels had worse accuracy on clean examples, likely because there was not enough information to correctly classify the image, but greater robustness to adversarial examples, likely because larger buckets mean a greater chance that a given perturbation will not yield any change in input to the network (Xu et al., 2017). Results can be seen in Tables 12, and are visualized in Figure 4.

| | | | White-box | Black-box | |
|---|---|---|---|---|---|
| **Levels** | **Clean** | | **PGD/LS-PGA** | **Vanilla, Clean** | **Vanilla, PGD** |
| Vanilla (Madry) | 87.3 | | 50.00 | 85.60 | 67.00 |
| One-hot (4) | 83.67 | | 59.59 | 83.03 | 72.03 |
| One-hot (8) | 85.62 | | 61.98 | 84.92 | 72.10 |
| One-hot (16) | 88.54 | | 67.83 | 86.01 | 61.92 |
| One-hot (32) | 88.56 | | 67.82 | 85.69 | 66.64 |
| One-hot (64) | 89.63 | | 65.63 | 86.94 | 65.77 |
| Thermometer (4) | 84.47 | | 61.88 | 83.64 | **72.96** |
| Thermometer (8) | 85.17 | | 67.23 | 83.41 | 71.11 |
| Thermometer (16) | 89.88 | | **79.16** | **88.25** | 67.96 |
| Thermometer (32) | **90.30** | | 72.91 | 86.06 | 59.32 |
| Thermometer (64) | 89.95 | | 69.37 | 83.85 | 51.82 |

Table 12: Comparison of adversarial robustness on CIFAR-10 as the number of levels of discretization is varied. All models are trained *only* on adversarial examples.

# E    EXPERIMENTS ON CIFAR-100

We list the experimental results on CIFAR-100 in Table 13. We choose $\xi = 0.01$ and $\delta = 1.2$ for the LS-PGA attack hyperparameters. For the discretized models, we used 16 levels. All adversarially trained models were trained on a *mix* of clean and adversarial examples.

| | Source / Target | | White-box | Black-box | |
|---|---|---|---|---|---|
| | | **Clean** | **PGD/LS-PGA** | **Vanilla, Clean-trained** | **Vanilla, PGD-trained** |
| *Clean* | Vanilla | 74.32 | 0 | 0.4 | 9.40 |
| | One-hot | 73.25 | 16.55 | 55.72 | 19.33 |
| | Thermometer | **74.44** | 16.50 | 50.33 | 18.49 |
| *Adv.* | Vanilla | 64.46 | 6.02 | 7.46 | 11.77 |
| | One-hot | 66.54 | 25.11 | 63.09 | **42.46** |
| | Thermometer | 68.44 | **28.14** | **63.21** | 41.97 |

Table 13: Comparison of adversarial robustness on CIFAR-100. All adversarially trained models were trained on a *mix* of clean and adversarial examples.

# F    EXPERIMENTS ON SVHN

We list the experimental results on SVHN in Table 14. All adversarially trained models were trained *only* on adversarial examples.

| | Source | Clean | White-box | Black-box | |
| :--- | :--- | :---: | :---: | :---: | :---: |
| Target | | | **PGD/LS-PGA** | **Vanilla, Clean-trained** | **Vanilla, PGD-trained** |
| *Clean* | Vanilla | 97.90 | 6.99 | 73.94 | 42.04 |
| | One-hot | 97.59 | 56.02 | 75.77 | 41.59 |
| | Thermometer | 97.87 | 56.37 | 78.04 | 41.69 |
| *Adv.* | Vanilla | 94.79 | 59.63 | 81.24 | 46.77 |
| | One-hot | 95.12 | 87.34 | 83.84 | 43.40 |
| | Thermometer | 97.74 | **94.77** | **84.97** | **48.67** |

Table 14: Comparison of adversarial robustness on SVHN.

## G  SUPPLEMENTARY FIGURES

In Figure 4 we plot the effect of increasing the levels of discretization for the MNIST and CIFAR-10 datasets.

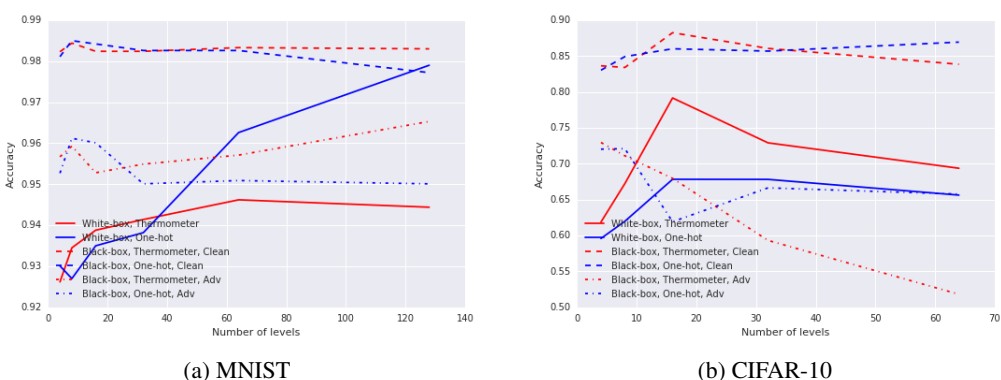

(a) MNIST            (b) CIFAR-10

Figure 4: The effect of increasing the number of distinct discretization levels on the accuracy of the model on MNIST and CIFAR-10. (4a) shows the accuracy on on MNIST for discretized models trained on a *mix* of legitimate and adversarial examples. (4b) shows the accuracy on CIFAR-10 for discretized models trained *only* on adversarial examples.

In Figure 5 we plot the convergence rate of clean trained and adversarially trained models on the CIFAR-10 dataset. Note that *thermometer encoded* inputs converge much faster in accuracy on both clean and adversarial inputs.

Figure 6 plots the norm of the gradient as a function of the number of iterations of the attack on MNIST. Note that the gradient vanishes at around 40 iterations, which coincides with the loss stabilizing in Figure 3.

In Figure 7, we create a linear interpolation between a clean image and an adversarial example, and then continue to extrapolate along this line, evaluating probability of each class at each point. In models trained on unquantized inputs, the class probabilities are all mostly piecewise linear in both the positive and negative directions. In contrast, the discretized model has a much more jagged and irregular shape.

In Figure 8, we plot the error for different models on various values of $\varepsilon$. The discretized models are extremely robust to all values less-than-or-equal-to the values that they have been exposed to during training. However, beyond this threshold, discretized models collapse immediately, while real-valued models still maintain some semblance of robustness. This exposes a weakness of the discretization approach; the same nonlinearity that helps it learn to become robust to all attacks it sees during training-time causes its behavior is unpredictable beyond that.

However, we believe that the fact that the performance of thermometer-encoded models degrades more quickly than that of vanilla models beyond the training epsilon is a significant weakness in practice, but no worse than other defenses. The "standard setting" for the adversarial example problem (in which we constrain the L-infinity norm of the perturbed image to an epsilon ball around the

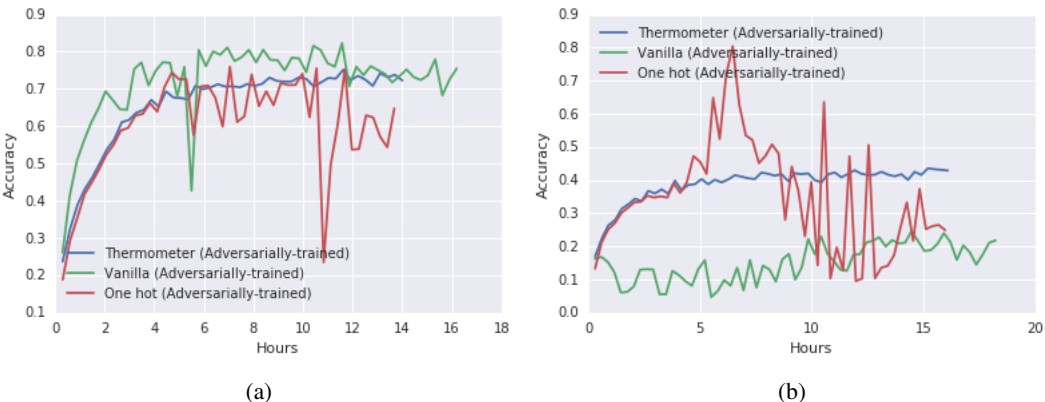

(a)                                          (b)

Figure 5: Comparison of the convergence rate of various *adversarially trained* models on the CIFAR-10 dataset. The discretized models use 16 levels per color channel. (5a) shows the accuracy on clean examples, while (5b) shows the accuracy on white-box PGD/LS-PGA examples, in wall-clock time.

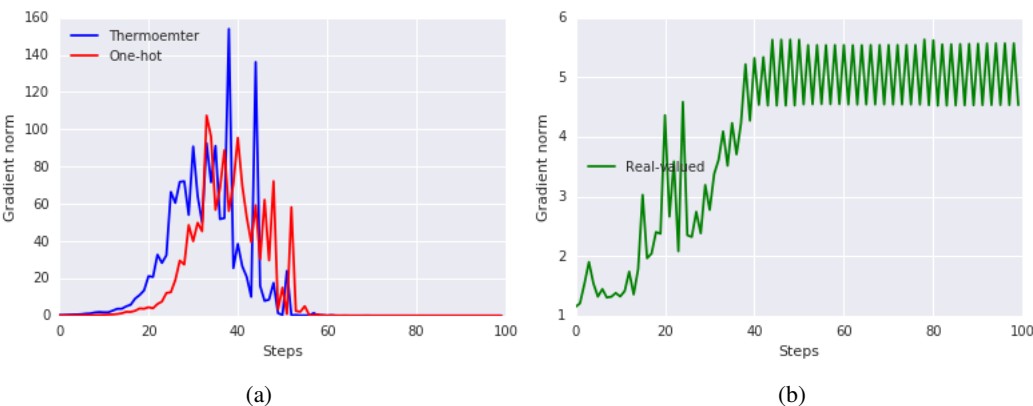

(a)                                          (b)

Figure 6: Gradient norm for iterated white-box attacks on various models on a randomly chosen data point from MNIST. (6a) shows the gradient norm on discretized models as a function of steps of LS-PGA, while (6b) shows the gradient norm on a vanilla model as a function of steps of PGD.

original image) was designed to ensure that any adversarially-perturbed image is still recognizable as its original image by a human. However, this artificial constraint excludes many other potential attacks that also result in human-recognizable images. State-of-the art defenses in the standard setting can still be easily defeated by non-standard attacks; for an example of this, see appendix A of ICLR submission "Adversarial Spheres". A "larger epsilon" attack is just one special case of a "non-standard" attack. If we permit non-standard attacks, a fair comparison would show that all current approaches are easily breakable. There is nothing special about the "larger epsilon" attack that makes a vulnerability to this non-standard attack in particular more problematic than vulnerabilities to other non-standard attacks.

In Figures 9, 10 , 11, 12 , 13 and 14 we plot several examples of church-window plots for MNIST (Goodfellow et al., 2014). Each plot is crafted by taking several test-set images, calculating the vector corresponding to an adversarial attack on each image, and then choosing an additional random orthogonal direction. In each plot, the clean image is at the center and corresponds to the color *white*, the $x$-axis corresponds to the magnitude of a perturbation in the adversarial direction, and the $y$-axis corresponds to the magnitude of a perturbation in the orthogonal direction. Note that we use the same random seed to generate the test set examples and the adversarial directions across different church-window plots.

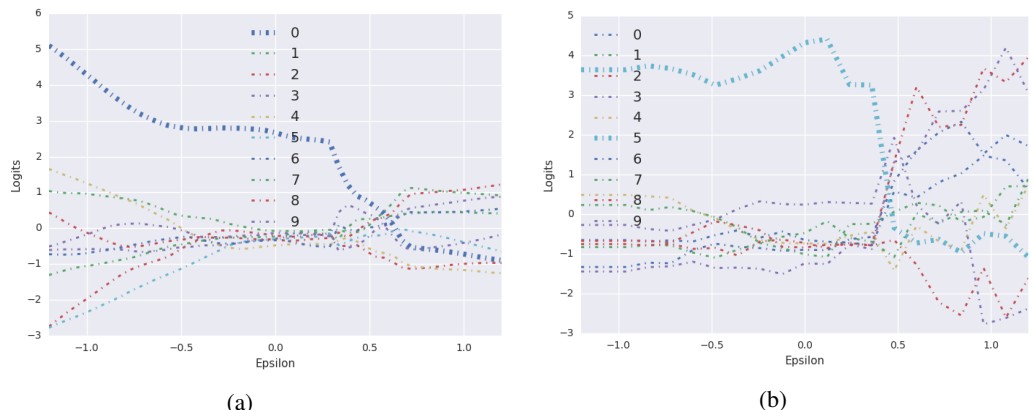

Figure 7: Linear extrapolation plot as in Goodfellow et al. (2014) for MNIST. (7a) shows the behavior of a vanilla model while (7b) for a discretized model using 16 levels and thermometer encoding. The $\varepsilon$-bound is $[-1.5, 1.5]$.

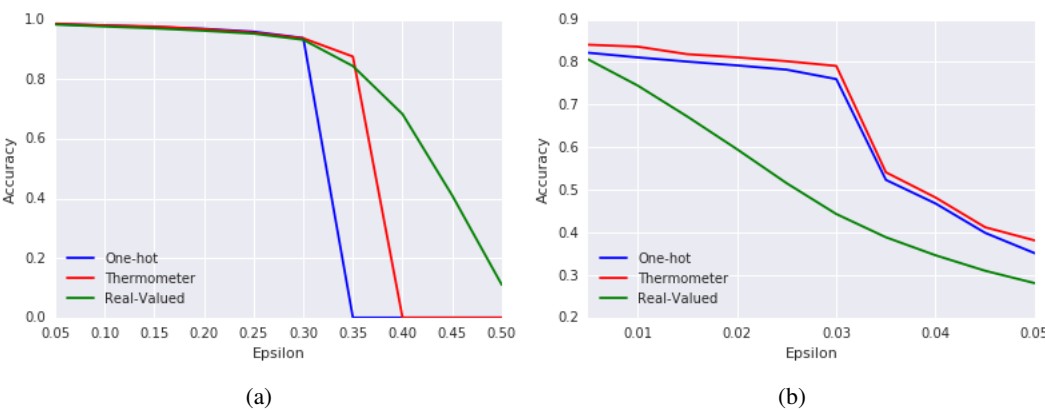

Figure 8: Plot showing the accuracy of various adversarially trained models on MNIST with $\varepsilon = 0.3$ (8a) and on CIFAR-10 with $\varepsilon = 0.031$ (8b), when attacked with increasing values of $\varepsilon$ using PGD/LS-PGA.

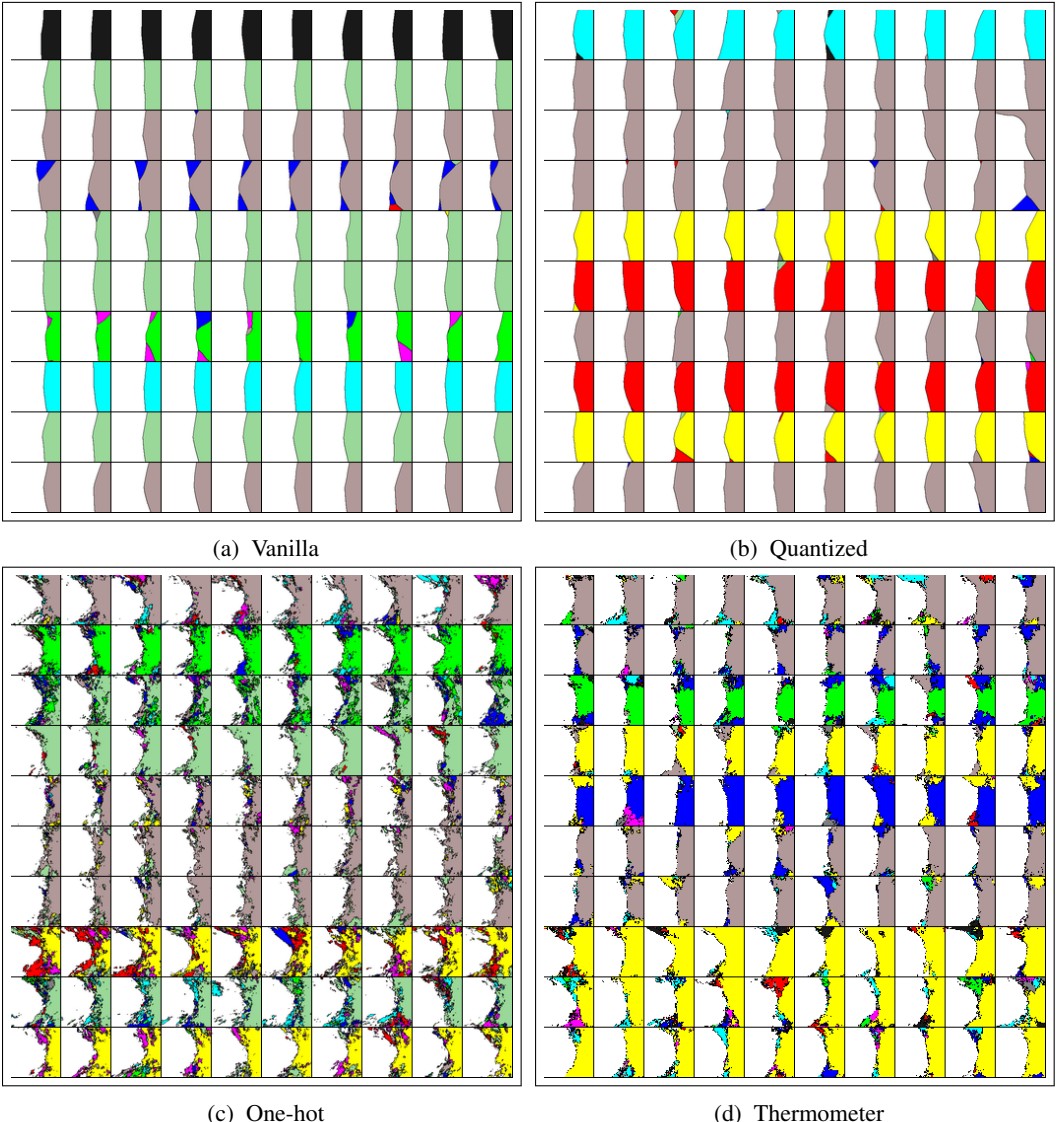

(a) Vanilla

(b) Quantized

(c) One-hot

(d) Thermometer

Figure 9: Church-window plots of clean-trained models on MNIST. The $x$-axis of each sub-plot represents the adversarial direction, while the $y$-axis represents a random orthogonal direction. The correct class is represented by *white*. Every row in the plot contains a training data point chosen uniformly at random, while each column uses a different random orthogonal vector for the $y$-axis. The $\varepsilon$ bound for both axes is $[-1.0, 1.0]$. Notice the almost-linear decision boundaries on non-discretized models.

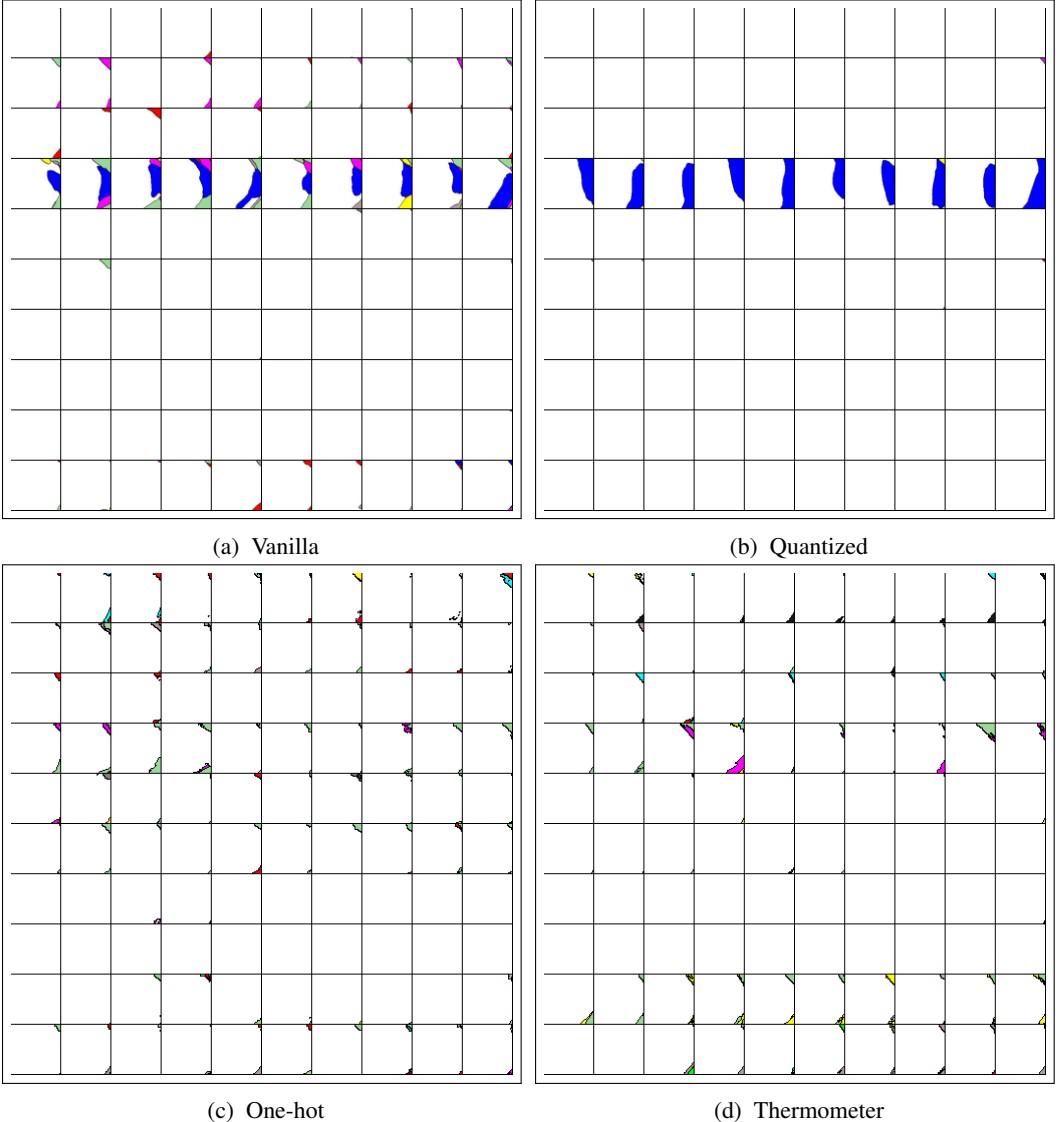

(a) Vanilla

(b) Quantized

(c) One-hot

(d) Thermometer

Figure 10: Church-window plots of adversarially-trained models on MNIST, trained on only adversarial examples. The $x$-axis of each sub-plot represents the adversarial direction, while the $y$-axis represents a random orthogonal direction. The correct class is represented by *white*. Every row in the plot contains a training data point chosen uniformly at random, while each column uses a different random orthogonal vector for the $y$-axis. The $\varepsilon$ bound for both axes is $[-1.0, 1.0]$.

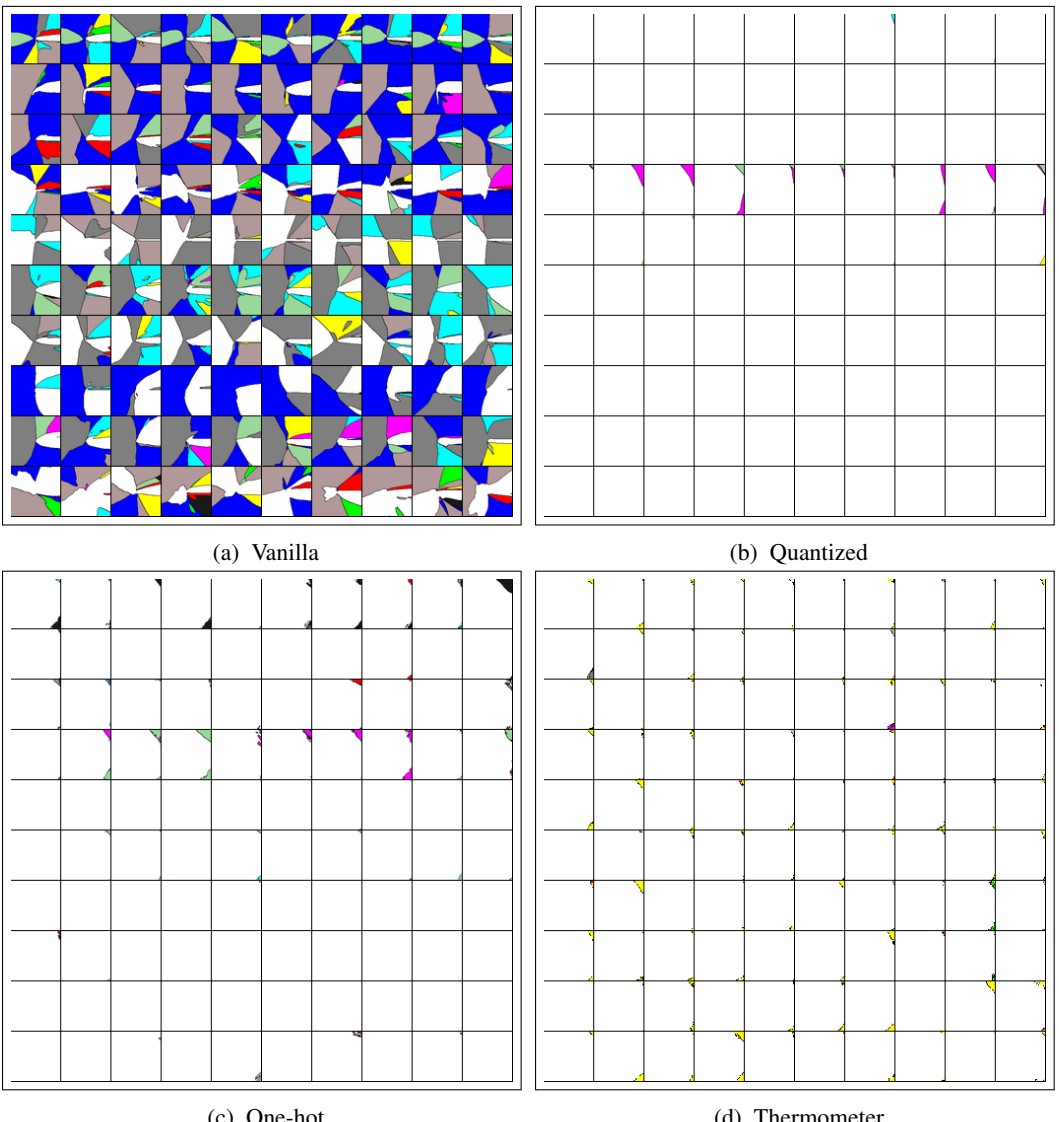

(a) Vanilla

(b) Quantized

(c) One-hot

(d) Thermometer

Figure 11: Church-window plots of adversarially-trained models on MNIST, trained using a mix of clean and adversarial examples. The $x$-axis of each sub-plot represents the adversarial direction, while the $y$-axis represents a random orthogonal direction. The correct class is represented by *white*. Every row in the plot contains a training data point chosen uniformly at random, while each column uses a different random orthogonal vector for the $y$-axis. The $\varepsilon$ bound for both axes is $[-1.0, 1.0]$. Notice the almost-linear decision boundaries on non-discretized models.

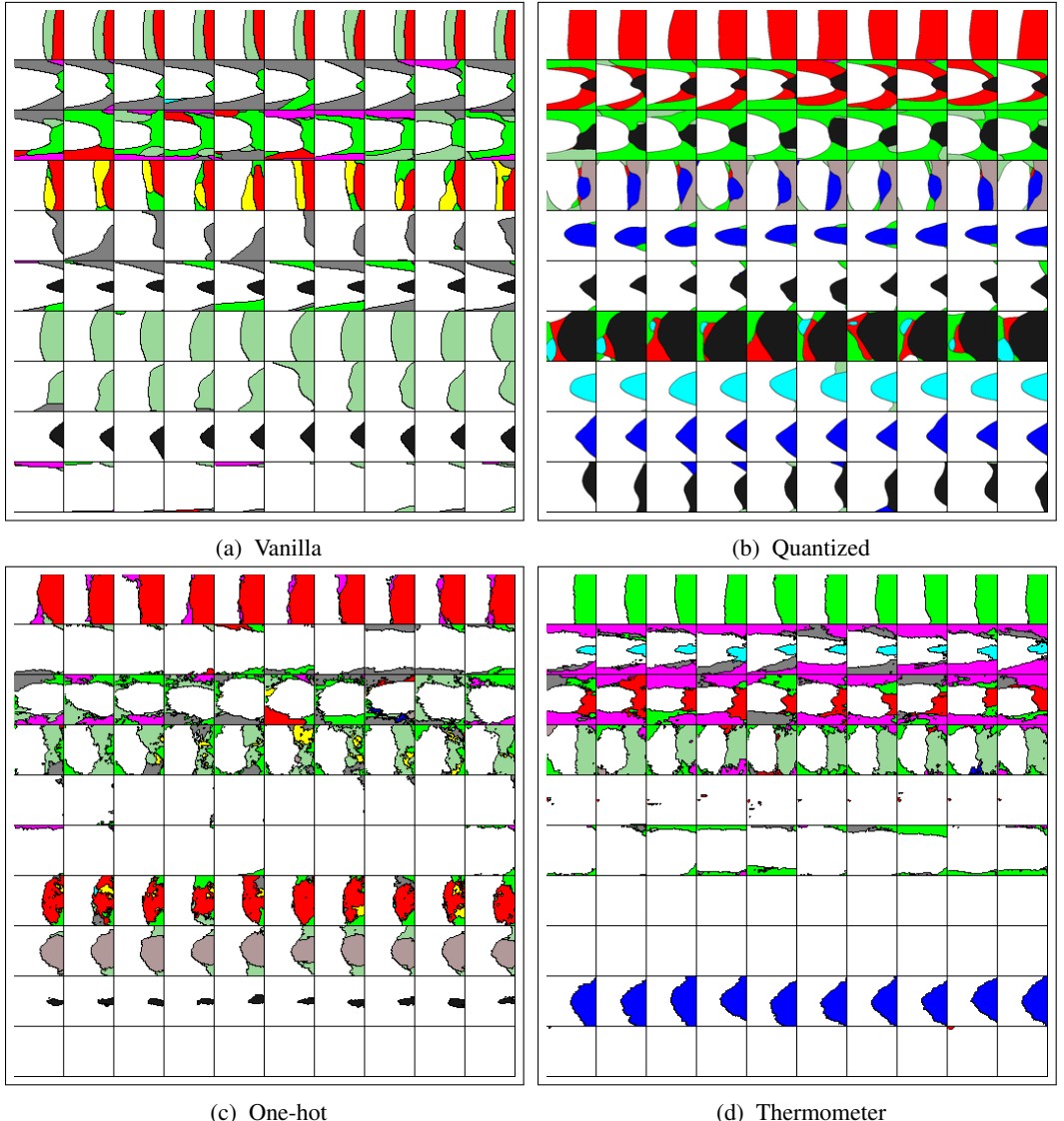

(a) Vanilla

(b) Quantized

(c) One-hot

(d) Thermometer

Figure 12: Church-window plots of clean-trained models on CIFAR-10. The $x$-axis of each sub-plot represents the adversarial direction, while the $y$-axis represents a random orthogonal direction. The correct class is represented by *white*. Every row in the plot contains a training data point chosen uniformly at random, while each column uses a different random orthogonal vector for the $y$-axis. The $\varepsilon$ bound for both axes is $[-1.0, 1.0]$. Notice the almost-linear decision boundaries on non-discretized models.

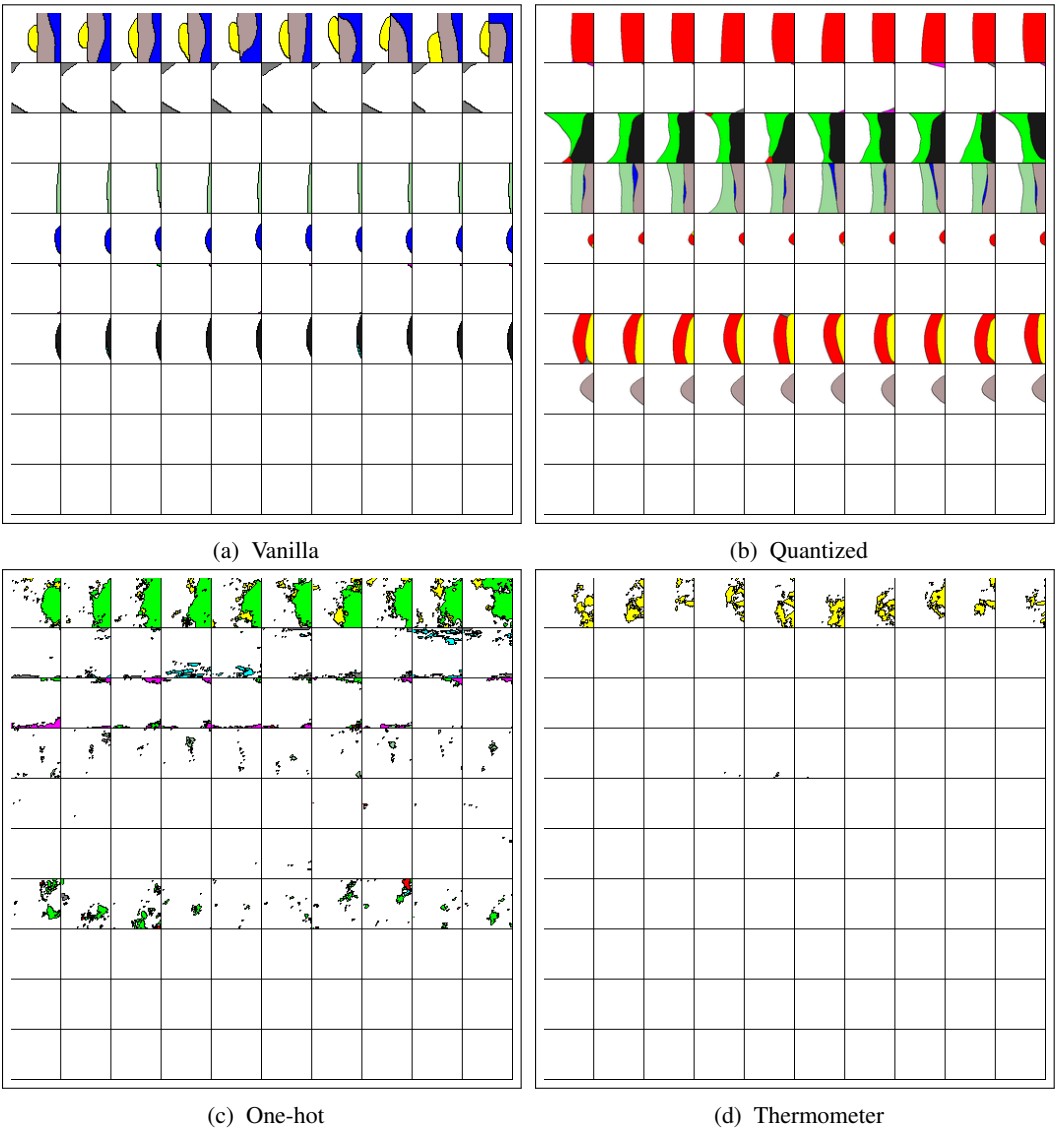

Figure 13: Church-window plots of adversarially-trained models on CIFAR-10, trained on only adversarial examples. The $x$-axis of each sub-plot represents the adversarial direction, while the $y$-axis represents a random orthogonal direction. The correct class is represented by *white*. Every row in the plot contains a training data point chosen uniformly at random, while each column uses a different random orthogonal vector for the $y$-axis. The $\varepsilon$ bound for both axes is $[-1.0, 1.0]$.

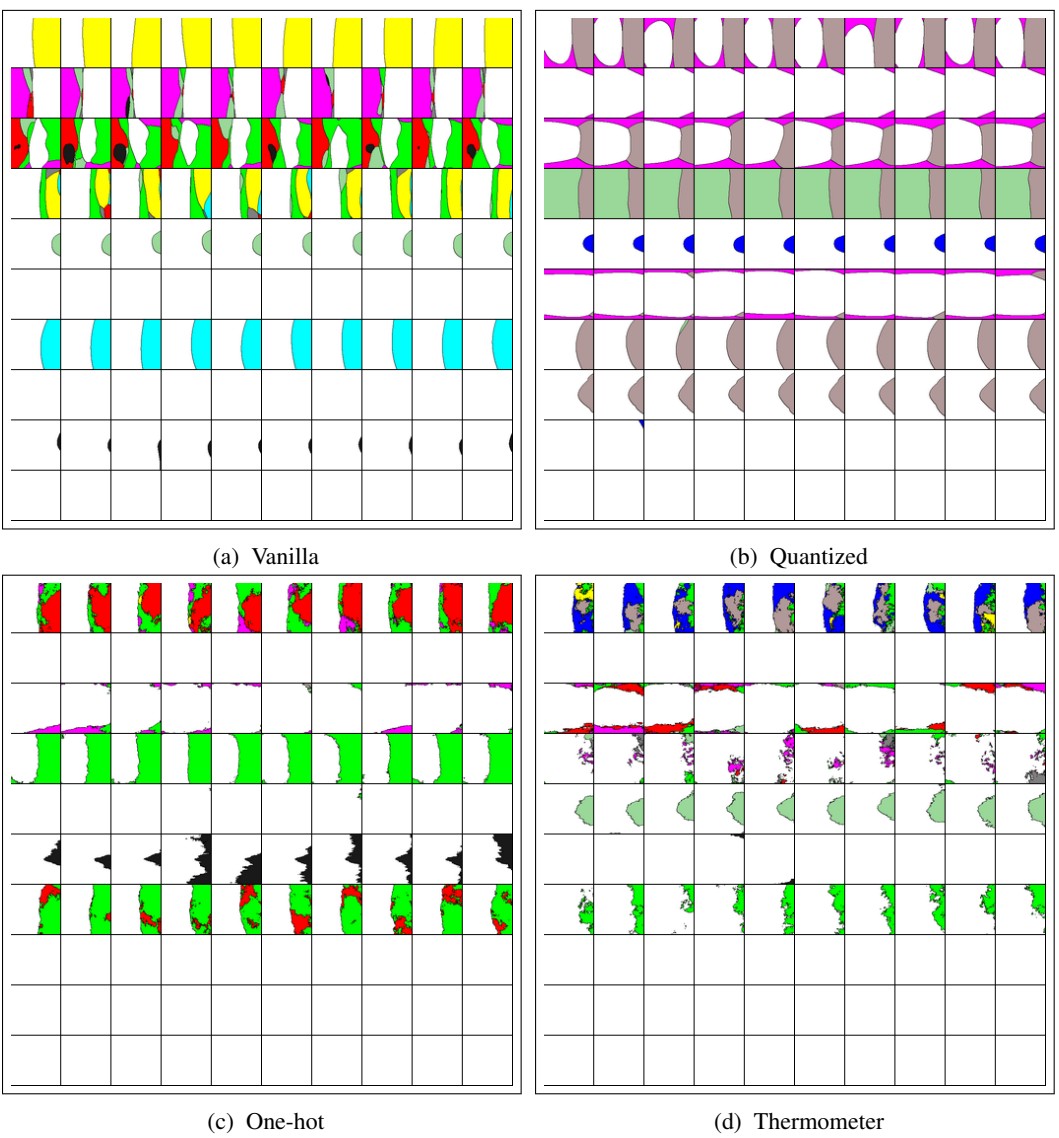

(a) Vanilla

(b) Quantized

(c) One-hot

(d) Thermometer

Figure 14: Church-window plots of adversarially-trained models on CIFAR-10, trained using a mix of clean and adversarial examples. The $x$-axis of each sub-plot represents the adversarial direction, while the $y$-axis represents a random orthogonal direction. The correct class is represented by *white*. Every row in the plot contains a training data point chosen uniformly at random, while each column uses a different random orthogonal vector for the $y$-axis. The $\varepsilon$ bound for both axes is $[-1.0, 1.0]$. Notice the almost-linear decision boundaries on non-discretized models.

