# OpenReview forum: "Thermometer Encoding: One Hot Way To Resist Adversarial Examples"
_ICLR.cc/2018/Conference — Accept (Poster)_

### Official Review · AnonReviewer1 · 2017-11-25
**Thermometer encoding is an interesting input discretization that is empirically shown to be robust to adversarial examples.**

**Rating:** 6
**Confidence:** 2

**Review:**

This paper studies input discretization and white-box attacks on it to make deep networks robust to adversarial examples. They propose one-hot and thermometer encodings as input discretization and
also propose DGA and LS-PGA as white-box attacks on it.
Robustness to adversarial examples for thermometer encoding is demonstrated through experiments.

The empirical fact that thermometer encoding is more robust to adversarial examples than one-hot encoding,
is interesting. The reason why thermometer performs better than one-hot should be pursued more.

[Strong points]
* Propose a new type of input discretization called thermometer encodings.
* Propose new white-box attacks on discretized inputs.
* Deep networks with thermometer encoded inputs empirically have higher accuracy on adversarial examples.

[Weak points]
* No theoretical guarantee for thermometer encoding inputs.
* The reason why thermometer performs better than one-hot has not unveiled yet.

[Detailed comments]
Thermometer encodings do not preserve pairwise distance information.
Consider the case with b_1=0.1, b_2=0.2, b_3=0.3, b_4=0.4 and x_i=0.09, x_j=0.21 and x_k=0.39.
Then, 0.12=|x_j-x_i|<|x_k-x_j|=0.18 but ||tau(b(x_i))-tau(n(x_j))||_2=sqrt(2)>1=||tau(b(x_k))-tau(n(x_j))||_2.

---

> ### Author Response · Authors · 2018-01-05
> **Empirical results likely explainable due to inductive bias**
>
> Thank you for your feedback! The finding that thermometer encodings perform better than one-hot encodings was a primarily empirical finding, and further exploration is certainly needed. The goal of this work was primarily to establish the strength of discretized encodings, and thermometer encodings in particular. We currently believe that the primary reason that thermometer encodings perform better and exhibit smoother convergence than one-hot encodings in the adversarial defense regime is the superior inductive bias; nearby pixel values typically have similar semantic content.
>
> You are correct that thermometer encodings do not preserve pairwise distance information; our statement was true only in the case where the number of discretization levels is equal to the number of possible pixel values, as in the PixelRNN paper. Since this is not realistic in most settings, we have weakened our claim to instead state that thermometer encodings maintain ordering information, which is true in all cases.

---

### Official Review · AnonReviewer2 · 2017-11-29
**Interesting, I want more.**

**Rating:** 6
**Confidence:** 4

**Review:**

This is a beautiful work that introduces both (1) a novel way of defending against adversarial examples generated in a black-box or white-box setting, and (2) a principled attack to test the robustness of defenses based on discretized input domains. Using a binary encoding of the input to reduce the attack surface is a brilliant idea. Even though the dimensionality of the input space is increased, the intrinsic dimensionality of the data is drastically reduced. The direct relationship between robustness to adversarial examples and intrinsic dimensionality is well known (paper by Fawzi.). This article exploits this property nicely by designing an encoding that preserves pairwise distances by construction. It is well written overall, and the experiments support the claims of the authors.

This work has a crucial limitation: scalability.
The proposed method scales the input space dimension linearly with the number of discretization steps. Consequently, it has a significant impact on the number of parameters of the model when the dimensionality of the inputs is large. All the experiments in the paper report use relatively small dimensional datasets. For larger input spaces such as Imagenet, the picture could be entirely different:

	- How would thermometer encoding impact the performance on clean examples for larger dimensionality data (e.g., Imagenet)?
	- Would the proposed method be significantly different from bit depth reduction in such setting?
	- What would be the impact of the hyper-parameter k in such configuration?
	- Would the proposed method still be robust to white box attack?
	- The DGA and LS-PGA attacks look at all buckets that are
within ε of the actual value, at every step. Would this be feasible in a large dimensional setting? More generally, would the resulting adversarial training technique be practically possible?

While positive results on Imagenet would make this work a home run,  negative results would not affect the beauty of the proposal and would shed critical light on the settings in which thermometer encoding is applicable. I lean on the accept side, and I am willing to increase the score greatly if the above questions are answered.

---

> ### Author Response · Authors · 2018-01-05
> **Scaling seems feasible, but no empirical results yet**
>
> Thank you for your feedback! We agree that learning more about the scaling properties would be enormously useful, but unfortunately, running adversarial training on full-size ImageNet simply proved too challenging. In our first attempt, we were forced to reduce our batch size in order to fit everything into memory, but this lead to poor convergence. We have now scaled up our resources in order to run it properly, and we hope to have results within the next few weeks; however, these experiments are still ongoing, and we have no results to report at this time. Unfortunately, since most of your concerns are empirical, this means that we cannot properly address them.
>
> One note is that when setting up these experiments, the primary bottleneck was the memory consumption of the model itself, especially under multiple steps of attack. It’s true that increasing the number of discretization levels causes a linear increase in memory consumption proportional to the size of the input, but this is negligible compared to the memory usage of the actual model: in our Wide ResNet implementation, we estimate the memory used by the first layer (and thus multiplied by the input discretization step) to be only 1-2% of the overall memory used. Both vanilla and 16-level-thermometer-encoded inputs were subject to approximately the same constraints on batch sizes, steps of attack, etc., when using a wide ResNet with width of 4 and depth of 16.
>
> In addition to having a relatively small proportional memory increase, input discretization requires relatively few additional model parameters: 0.03% extra parameters for MNIST, 0.08% for CIFAR-10 and CIFAR-100, and 2.3% for SVHN, as described in section 5. This indicates that the model size will not be a bottleneck in scaling discretized models.

---

### Official Review · AnonReviewer3 · 2017-12-04
**an interesting study, but the validity of the approach is still to be demonstrated**

**Rating:** 6
**Confidence:** 4

**Review:**

The authors present an in-depth study of discretizing / quantizing the input as a defense against adversarial examples. The idea is that the threshold effects of discretization make it harder to find adversarial examples that only make small alterations of the image, but also that it introduces more non-linearities, which might increase robustness. In addition, discretization has little negative impact on the performance on clean data. The authors also propose a version of single-step or multi-step attacks against models that use discretized inputs, and present extensive experiments on MNIST, CIFAR-10, CIFAR-100 and SVHN, against standard baselines and, on MNIST and CIFAR-10, against a version of quantization in which the values are represented by a small number of bits.

The merits of the paper is that the study is rather comprehensive: a large number of datasets were used, two types of discretization were tried, and the authors propose an attack mechanism better that seems reasonable considering the defense they consider. The two main claims of the paper, namely that discretization doesn't hurt performance on natural test examples and that better robustness (in the author's experimental setup) is achieved through the discretized encoding, are properly backed up by the experiments.

Yet, the applicability of the method in practice is still to be demonstrated. The threshold effects might imply that small perturbations of the input (in the l_infty sense) will not have a large effect on their discritized version, but it may also go the other way: an opponent might be able to greatly change the discretized input without drastically changing the input. Figure 8 in the appendix is a bit worrysome on that point, as the performance of the discretized version drops rapidly to 0 when the opponents gets a bit stronger. Did the authors observe the same kind of bahavior on other datasets? What would the authors propose to mitigate this issue? To what extend the good results that are exhibited in the paper are valid over the wide range of opponent's strengths?

minor comment:
- the experiments on CIFAR-100 in Appendix E are carried out by mixing adversarial / clean examples while training, whereas those on SVHN in Appendix F use adversarial examples only.

---

> ### Author Response · Authors · 2018-01-05
> **Performance drop of discretized models past training threshold is not a weakness in practice**
>
> Thank you for your feedback!
>
> > ...the performance of the discretized version drops rapidly to 0 when the opponents gets a bit stronger.
>
> We observe this behavior on the datasets we tested on, which were MNIST and CIFAR. (CIFAR does not drop all the way to 0, but does drop sharply.) We believe that this is an expected result, stemming from the intuition that the relationship between the input and the loss is highly nonlinear. When presented with an input which it has never been exposed to (i.e. a pixel has been moved into a bucket that is beyond the adversarial training threshold), the effect on the loss is highly random. Many of these perturbed inputs will increase the loss, and it is therefore easy to find an adversarial example.
>
> Controlling for the wide range of opponent’s strengths is an important issue, one which is endemic to adversarial defenses in general. The “standard setting” for the adversarial example problem (in which we constrain the L-infinity norm of the perturbed image to an epsilon ball around the original image) was designed to ensure that any adversarially-perturbed image is still recognizable as its original image by a human. However, this artificial constraint excludes many other potential attacks that also result in human-recognizable images. State-of-the art defenses in the standard setting can still be easily defeated by non-standard attacks; for recent examples of this, see ICLR submission “Adversarial Spheres” (appendix A), as well as “Adversarial Patch” by Brown et. al (https://arxiv.org/abs/1712.09665).
>
> With this in mind, we believe that the fact that the performance of thermometer-encoded models degrades more quickly than that of vanilla models beyond the training epsilon is a weakness, but no worse in practice than other defenses. A “larger epsilon” attack is just one special case of a “non-standard” attack; there are an enormous number of other non-standard attacks, some of which are more effective against vanilla models, some of which are more effective against thermometer encodings, and some of which are devastating to both. If we permit non-standard attacks, a fair comparison would show that all current approaches are easily breakable. There is nothing special about the “larger epsilon” attack that makes a vulnerability to this non-standard attack in particular more problematic than vulnerabilities to other non-standard attacks, in practice.
>
> Additionally, on the CIFAR dataset, we found that even though discretized inputs are impacted much more severely by examples perturbed by more than the training threshold, the discretized models are sufficiently strong to begin with that they still outperform real-valued models even after this vulnerability has been exploited. (See updated Figure 8b.) CIFAR is more reflective of real-world datasets, so even with this weakness, thermometer-encoded models may outperform real-valued models in practice.
>
> Based on your feedback, we updated our submission to include this discussion in Appendix G, and added Figure 8b showing the CIFAR results. Also, we discovered a bug which caused the unquantized attack in Figure 8 to be too weak; essentially, we were using a fixed step size of 0.01 for 40 steps which caused the perturbation to never hit the boundary for epsilon > 0.4. We have updated the figure to reflect the correct values. (The fixed results are qualitatively equivalent, so this change does not affect the conclusions.)

---

### Public Comment · ~Micah_Sheller1 · 2017-11-02
**Appendix F - SVHN results**

As I understand these results, I think that the SVHN results in Table 14 are very curious, and would like to see more analysis there. The discrepancy between white-box and black-box is quite odd, as are the limited gains of PGD-trained themometer target model.

---

> ### Author Response · Authors · 2017-11-16
> **Thanks**
>
> Thanks very much for your question and interest in our paper.
>
> We do not know for sure what causes the higher attack success rate for black-box adversarial examples compared to white box adversarial examples in Table 14. This discrepancy is consistent with the commonly-observed "gradient masking" problem that can be largely overcome with ensemble adversarial training: https://arxiv.org/pdf/1705.07204.pdf
> It is possible that combining thermometer encoding and ensemble adversarial training could yield models that retain the white box robustness we have obtained with thermometer encoding but also have increased black box robustness.
> However, we have not done enough tests to verify that the issue is gradient masking, so we can't guarantee that ensemble adversarial training would help. This investigation is left to future work.
>
> We don't understand what you mean by the "limited gains of PGD-trained themometer target model."
> The thermometer model is trained using LS-PGA adversarial examples, not PGD adversarial examples, but we interpret your comment to mean adversarially-trained thermometer models. We aren't sure whether you mean that adversarial training gives limited improvement to thermometer models or that thermometer models give limited improvement to adversarial training. Neither of these is supported by Table 14. Adversarial training causes thermometer models to become state of the art in all three categories in table 14. Likewise, thermometer coding causes adversarial training to become state of the art in all three categories. If you're concerned that the difference caused by thermometer coding on black box adversarial examples is small enough that it might be statistically insignificant, we can add error bars showing the 95% confidence interval. We can tell you ahead of time that these error bars do not overlap. SVHN has over 26,000 test examples, so the standard error of the test accuracy is smaller than on datasets with smaller test sets like MNIST and CIFAR.
>
> Again, thank you for your interest.

---

> > ### Public Comment · ~Micah_Sheller1 · 2017-11-27
> > **Apologies for confusion**
> >
> > Apologies for my mix-up on the training. I meant the relatively small difference between the Clean and LS-PGA targets when attacked by the Blackbox PGD method. I hope that clarifies. It's really the same issue: the LS-PGA trained target performs poorly against the Blackbox PGD attacker.
> >
> > I'll read the paper you've linked. Thanks!

---

### Public Comment · (anonymous) · 2017-11-22
**Beautiful!**

Nice work, do you reckon that the density of adv. samples has gone down compared to Madry et al., or is it just that they are hard to find using gradient based techniques?

---

### Decision · Program_Chairs · 2018-01-29
**ICLR 2018 Conference Acceptance Decision**

**Decision:**

Accept (Poster)

**Comment:**

This paper is borderline.  The reviewers agree that the method is novel and interesting, but have concerns about scalability and weakness to attacks with larger epsilon.  I will recommend accepting; but I think the paper would be well served by imagenet experiments, and hope the authors are able to include these for the final version